# Early Dirty Buffer Flush with Second Chance for SSDs

**DOI:** 10.3390/mi14040796

**Published:** 2023-03-31

**Authors:** Ilhoon Shin

**Affiliations:** Department of Electronic Engineering, Seoul National University of Science and Technology, Seoul 01811, Republic of Korea; ilhoon.shin@seoultech.ac.kr; Tel.: +82-2-970-6420

**Keywords:** SSD, buffer cache, NVRAM, NAND flash memory, early flush, second chance

## Abstract

As high-performance server-based applications become more prevalent, there is a growing demand for high-performance storage solutions. In response, SSDs that use NAND flash memory as storage media are quickly replacing hard disks in the high-performance storage market. One way to improve SSD performance is to use an internal large-capacity memory as a buffer cache for NAND. Previous studies have shown that early flushing, which ensures sufficient clean buffers by flushing dirty buffers to NAND in advance when the ratio of dirty buffers exceeds a threshold, significantly reduces the average response time of I/O requests. However, the early flush can also have a negative side effect, namely an increase in NAND write operations. To address this problem, this study proposes a selective early flush policy. This policy evaluates the likelihood of a candidate dirty buffer being rewritten upon the early flush, and delays flushing if the candidate has a high rewrite likelihood. Through this selective early flush, the proposed policy reduces NAND write operations by up to 18.0% compared to the existing early flush policy in the mixed trace. Additionally, the response time of I/O requests is also improved in most of the considered configurations.

## 1. Introduction

Recently, the use of server-based applications, such as cloud computing, virtualization, big data analysis, and social network services, has increased rapidly, driving a corresponding increase in demand for high-capacity and high-performance storage solutions that can quickly store and retrieve large amounts of data. NAND-based solid-state drives (SSDs) have achieved large capacities comparable to those of hard disks, thanks to the scaling down of semiconductor fabrication processes and the development of NAND chip-manufacturing technologies such as 3D TLC and 3D QLC [1,2,3]. In addition, SSDs provide much higher performance than hard disks by processing I/O requests in parallel at various levels, such as channel striping, die interleaving, and multi-plane [4,5,6,7,8,9,10], and by deploying large-capacity non-volatile random-access memory (NVRAM) or dynamic random-access memory (DRAM) inside the SSD and using it as a buffer cache for NAND [11,12,13,14,15,16,17,18,19,20]. As a result, it is now commonplace to build storage solutions using only SSDs in various server environments that require high-performance storage solutions [21,22,23,24].

When using large-capacity memory as a buffer cache, it is necessary to increase the hit ratio or reduce the miss penalty or hit time to achieve a short response time, as shown in Equation (1).
response time = HitRatio ∗ HitTime + (1 − HitRatio) ∗ MissPenalty(1)

Since hardware performance mainly determines the hit time, most studies have aimed to increase the cache hit ratio by using efficient replacement policies [11,12,13,14,15,16,17,18,19]. However, not much attention has been paid to reducing the miss penalty, which can be improved through software. The latency of the miss penalty varies significantly depending on whether the victim buffer is dirty or not during buffer replacement [20,25]. For instance, if the replaced buffer contains modified data and is in a dirty state, its data must first be written to NAND, resulting in an added flush time to the miss penalty. Therefore, to minimize the miss penalty, a clean buffer should be chosen as the victim buffer during buffer replacement.

Most SSD buffer replacement policies are based on the Clean First LRU (CFLRU) policy, which preferentially replaces clean buffers rather than dirty ones [14,15,16,17,18]. As a result, their initial miss penalty does not include the buffer flush time. However, replacing clean buffers continuously leads to a lack of clean buffers, and eventually, dirty buffers are replaced. To address this issue, we proposed an early flush policy in a previous study that flushes dirty buffers to NAND during idle time when the dirty buffer ratio exceeds a threshold, ensuring a sufficient number of clean buffers [25]. The early flush can be applied to any buffer replacement policy that prioritizes replacing clean buffers, resulting in a significantly shorter average response time for I/O requests by reducing the miss penalty. However, it has a downside: the total count of NAND write operations increases. If an early-flushed buffer becomes dirty again due to a write request from a host, the corresponding buffer will need to be written again to NAND later, resulting in an additional NAND write due to the early flush. This increase in NAND writes leads to more frequent garbage collection, which in turn increases the number of NAND block erases. If NAND blocks are erased excessively, they can become unstable and cause long-term damage to the lifespan of SSDs [3,24]. The previous study [25] left this issue for future research, which is the main problem that this study aims to address.

To prevent an increase in the count of NAND write operations caused by the early flush, the target for the early flush should be a dirty buffer with a low likelihood of being rewritten. This likelihood can be assessed based on temporal locality, a well-observed data access pattern in various domains, including operating systems [26,27,28]. In other words, data that have not been accessed for a long time or have been accessed for the first time are unlikely to be used again in the future. On the other hand, data that have been recently accessed again within a certain time frame are likely to be accessed again in the near future. To apply this concept, a write count is maintained for each buffer. When performing the early flush, if the write count for a candidate dirty buffer is 1, indicating that it has not been rewritten for a certain period of time, the rewrite likelihood is considered low and the early flush is carried out immediately. Conversely, if the write count for the candidate is greater than 1, indicating that it has been rewritten during a certain period of time, its rewrite likelihood is considered high, and the early flush is postponed, giving it a second chance. This selective early flush minimizes the side effect of the increase in NAND write operations.

For the proposed policy to be used in SSDs, the overhead should not be excessive. The proposed policy is designed using three linked lists: a clean LRU list, a dirty LRU list, and a second-chance dirty LRU list to achieve a time complexity of O(1) for buffer hits and buffer misses. The downside is that it requires additional memory space to store the write count, which can range from 1 to 4 bytes per buffer. However, this memory overhead is not significant. If the capacity of the buffer cache is 1 GB and each buffer is 8 KB in size, the additional memory space required is between 128 and 512 KB. The performance evaluation of the proposed policy is performed by taking this memory overhead into account. The cache capacity is set to be smaller than that of the other policies by the amount of the memory overhead, but it still provides better performance.

Another limitation of the previous study [25] is that the effect of the early flush was evaluated only in an environment where NVRAM or battery-packed DRAM was used as the buffer cache. No forced buffer flush was performed except in the case of dirty buffers being replaced and flushed to NAND. However, applications running on servers ensure data dependencies by performing sync() requests, and operating systems typically flush dirty buffers periodically. For example, Ubuntu Linux flushes dirty buffers every 30 s by default. When using DRAM as a buffer cache, all dirty buffers inside the SSD must be flushed to NAND when sync() requests or periodic dirty buffer flushes are issued. Therefore, even if the early flush is not performed, clean buffers are reproduced, and the advantage of the early flush may be reduced, and only the disadvantage of increasing the count of write operations can be emphasized. We therefore evaluate the performance of the proposed policy even in an environment where all dirty buffers are flushed every 30 s and show that the early flush is still effective in reducing an average response time.

The contributions of this study can be summarized as follows. First, the proposed policy selects target buffers for the early flush based on their rewrite likelihood, resulting in mitigating the increase in the total count of NAND write operations caused by the early flush while also improving the average response time. Second, the proposed policy has a time complexity of O(1) for buffer hits and misses, and its memory overhead, which requires 1–4 bytes per buffer, is not excessive when compared to the performance gain. Third, the performance of the early flush is evaluated in two different environments: one in which forced buffer flushing is not performed, and another in which periodic dirty buffer flushing is forced.

The rest of the paper is organized as follows. Section 2 provides background information and related work. Section 3 describes the proposed policy and its design, and Section 4 presents the results of the performance evaluation. Finally, Section 5 concludes the paper.

## 2. Background and Related Work

SSDs use multiple NAND flash chips that are connected with parallel channels as storage media. Each NAND chip contains multiple dies that can perform parallel operations, and each die contains multiple planes. A plane is comprised of multiple NAND blocks, and each block contains multiple pages. Similar to a sector on a hard disk, a page is the basic unit of read/write operations. NAND is a type of EEPROM that does not support overwriting data. Data can only be written on clean pages, and pages on which data have been written are restored to a clean state via the erase operation of the block to which they belong. To support legacy file systems that are designed for hard disks with overwriting capability, SSDs include a flash translation layer (FTL) that emulates the overwriting with an out-of-place update [29,30,31,32]. During the out-of-place update, the valid location of data changes with every update, making it necessary to maintain a mapping table to remember the current location of data. As the mapping table needs to be referenced during every read/write operation, the entire table or frequently accessed parts of it need to be held in memory to prevent performance degradation. Therefore, SSDs contain processors to perform FTL and a large-capacity RAM to maintain the mapping table.

Several studies have been conducted to improve performance by utilizing part of an SSD’s internal RAM as a buffer cache for NAND [11,12,13,14,15,16,17,18,19]. Most of these studies aimed to increase the hit ratio of the buffer cache or generate a NAND-friendly write pattern that considers the physical characteristics of NAND. For example, policies such as FAB [11] and BPLRU [12] manage the buffer cache in units of NAND blocks and attempt to generate a large, sequential write pattern known to be NAND-friendly. However, nowadays, it is common for SSDs to divide large requests into multiple sub-requests for parallel processing, such as channel striping, die interleaving, or multiplane operations. As a result, the performance difference between sequential and random write patterns is no longer noticeable [8,9]. In other words, sequential write patterns are no longer more SSD friendly than random write patterns.

Furthermore, the relative cost of a write operation is higher than that of a read operation in NAND. Write operations typically take 10 times longer than reads, and accumulated write operations trigger garbage collection, which reclaims clean pages with erase operations. Garbage collection involves multiple page copies and block erases, making it a time-consuming job and a major cause of reduced SSD performance. Furthermore, as the number of block erases accumulates, the NAND block becomes too unstable to retain data due to accumulated damage to the oxide film surrounding the cell. For these reasons, most buffer cache policies for SSDs replace clean buffers before dirty buffers. For example, CFLRU [14] preferentially replaces clean buffers in the candidate pool and only replaces dirty buffers when no clean buffers are available in the pool. The size of the candidate pool is usually set to half or one third of the buffer cache. CCF-LRU [15], CLRU [16], and ADLRU [17] are based on CFLRU and classify the entire buffers into hot and cold according to their reference likelihood. These policies replace the cold clean buffer first when replacing the buffer. The PRLRU [18] policy also replaces clean cold buffers first in the candidate pool. All of these policies have the advantage of not including the dirty buffer flush time in the miss penalty by replacing the clean buffers initially. However, when clean buffers are exhausted, the time to flush dirty buffers is included in the miss penalty, increasing the response time of I/O requests.

To address the problem above, the previous study [25] proposed an early flush policy that secures sufficient clean buffers by flushing dirty buffers in advance during idle time when the ratio of dirty buffers exceeds a threshold. The early flush is applicable to all policies, such as CFLRU and PRLRU, that replace the clean buffer first. The early flush ensures that clean buffers are replaced in most cases, preventing the dirty buffer flush time from being included in the miss penalty and significantly reducing the average response time of I/O requests.

However, the early flush policy has the weakness of increasing the total count of NAND write operations. Upon early flush, it flushes the least recently used (LRU) dirty buffer that has not been referenced for the longest time without considering its rewrite possibility. If the flushed buffer becomes dirty again due to a host write, an additional buffer flush occurs later, increasing the count of NAND write operations. This increase in NAND write operations can damage the lifespan of the SSD in the long run. 

## 3. Early Flush with Second Chance

### 3.1. Overview

To avoid an increase in NAND write operations due to the early flush, it is important to evaluate the likelihood of dirty buffers being rewritten and only select buffers with a low rewrite probability for the early flush. The evaluation of the rewrite probability can be based on the observation of temporal locality, which is evident in various domains such as operating systems and databases. The temporal locality means that data that have been referenced recently are more likely to be referenced again in the future, whereas data that have not been referenced for a long time are less likely to be referenced again. Additionally, it is observed that re-referenced data are more likely to be referenced again than data that have been referenced for the first time [26,27,28].

Based on these observations, we propose a new policy called CFLRU-EF-SC (CFLRU with early flush and second chance), which incorporates a second chance mechanism into the CFLRU-EF policy proposed in our previous study [25]. The CFLRU-EF-SC policy manages clean and dirty buffers separately, replacing the least recently used (LRU) clean buffer first when replacing buffers. If there is no clean buffer, the LRU dirty buffer is replaced.

Similar to the CFLRU-EF policy, the early flush is performed during idle time to secure sufficient clean buffers in advance when the ratio of dirty buffers falls below the threshold. When performing the early flush, the LRU dirty buffer is selected as a candidate. If the candidate has no rewrite record for a certain period of time, it is immediately flushed to NAND. However, if the candidate has a rewrite record, it is considered probable to be rewritten and given a second chance rather than being flushed. When the second chance is given, the candidate’s rewrite record is cleared. Therefore, if this candidate is not rewritten and is again selected for the early flush, it will be flushed immediately. If it is rewritten, it will get another second chance.

### 3.2. Data Structures

The CFLRU-EF-SC policy uses three lists to manage the buffers: a clean LRU list, a dirty LRU list, and an SC (second chance) dirty LRU list. The clean LRU list maintains a linked list of clean buffers in reference order, while the dirty LRU list maintains a linked list of dirty buffers in reference order. The SC dirty LRU list links dirty buffers that were given a second chance upon the early flush. At system startup, all buffers are initially connected to the clean LRU list, and both the dirty LRU list and the SC dirty LRU list are empty.

Each buffer stores its write count and current position information, which indicates the list it is connected to. The write count is used to evaluate the likelihood of a rewrite during the early flush. At system startup, the write counts of all buffers are initialized to 0. 

### 3.3. Operations

When a read request arrives from a host, the policy first checks if the requested data are in the buffer. If the data are found in the buffer (read hit), they are transferred from the buffer to the host without performing a NAND operation. The buffer that contains the requested data is moved to the head position of the list it is linked to. For example, when the buffer Y(0,α) is referenced, it is moved to the head of the clean LRU list, as shown in Figure 1a, where 0 is the current write count of the buffer Y, and α means that the buffer Y is currently linked to the clean LRU list. If the data are not found in the buffer (read miss), the policy performs buffer replacement. If the victim buffer is clean, the requested data are read from NAND into the victim buffer immediately. However, if the victim buffer is dirty, it is first flushed to NAND, and then the requested data are read from NAND. The victim buffer is then moved to the head position of the clean list.

When a write request arrives from a host, the policy checks if the requested data are in the buffer. If the data are found in the buffer (write hit), they are transferred from the host to the buffer without performing a NAND operation. If the referenced buffer is clean (case (1) in Figure 1b), it is removed from the clean LRU list and moved to the head position of the dirty LRU list. As shown in Figure 1b, the buffer C(0,α) is changed to C(1,β), where the write count is increased by 1 and the location information is updated. The state of the buffer is also changed to dirty. If the referenced buffer is found in the dirty LRU list, it is moved to the head position of the dirty LRU list (case (2) in Figure 1b). The write count is increased by 1, and the location information is unmodified. Lastly, if the referenced buffer exists in the SC dirty LRU list, it is moved to the head position of the SC dirty LRU list (case (3) in Figure 1b). The write count is increased by 1, and the location information is unmodified.

If the buffer is not found (write miss), a buffer replacement is performed. If the victim buffer is dirty, it is flushed first, and then the data are written from the host to the buffer. The victim buffer is then moved to the head of the dirty LRU list. Its state is set to dirty and the write count is set to 1.

When a buffer needs to be replaced, the victim buffer is chosen from the clean LRU list first, then the dirty LRU list, and finally the SC dirty LRU list. If the clean LRU list is not empty, its tail buffer is replaced. Otherwise, the tail buffer of the dirty LRU list is chosen for replacement. As mentioned in the introduction, if a dirty buffer is replaced, the flush time is added to the miss penalty, leading to a longer average response time for requests.

To minimize this problem, an early flush is performed when the ratio of dirty buffers exceeds a threshold. During idle time, dirty buffers are flushed to NAND to secure enough clean buffers. The tail buffer of the dirty LRU list is chosen as the candidate for the early flush, as illustrated in Figure 2a. If the write count of the candidate is 1, it has not been rewritten for a certain period of time since its initial write, and is therefore unlikely to be rewritten. As a result, it is immediately flushed, and its state is changed to clean and moved to the head of the clean LRU list. In this scenario, the write count remains unchanged.

On the other hand, if the write count is 2 or more, the candidate is likely to be rewritten in the future because it has a rewrite history. Therefore, it is not flushed and is given a second chance. In Figure 2b, the candidate L(3,β) is not flushed and is moved to the head of the SC dirty LRU list. Its state remains dirty, but the write count is reset to 1, clearing the rewrite history. The next tail buffer, T(1,β), becomes the new candidate for the early flush, and the same process is repeated until enough clean buffers have been generated. To take advantage of the parallel processing capability of the SSD, the policy flushes as many dirty buffers as possible together, equal to the number of dies in a chip. 

Meanwhile, if the size of the dirty LRU list is significantly smaller than that of the SC dirty LRU list, there is a risk that the first written buffer will be selected as the target of the early flush too soon. Therefore, the proposed policy maintains the size of the dirty LRU list at least equivalent to that of the SC dirty LRU list. To achieve this, after completing the early flush, if the size of the SC dirty LRU list exceeds that of the dirty LRU list, the tail buffers of the SC dirty list are moved to the head of the dirty LRU list until the sizes of both lists are the same. Consequently, if buffers given a second chance are not rewritten subsequently, they will eventually be selected as targets for the early flush. The pseudo code of the early flush is described in Figure 3.

### 3.4. Dynamic Adjustment of a Threshold

The proposed policy adopts the threshold adjustment method described in our previous study [25]. At system startup, the threshold is set at a fixed value of 80%, and it is periodically adjusted after every 1000 host write requests, considering the number of dirty buffer replacements and the number of rewrites that occur after performing the early flush. If the number of dirty buffer replacements increases, the threshold is decreased to ensure that there are sufficient clean buffers available. On the other hand, if there are more rewrites in the early flushed buffers, the threshold is increased to avoid triggering the early flush too often. Refer to [25] for more detailed equations related to this adjustment method.

### 3.5. Overheads Analysis

In order for the proposed policy to be applicable to commercial SSDs, its computational and memory overhead should not be excessive when compared to existing policies such as CFLRU. The proposed policy has a time complexity of O(1) when handling foreground I/O requests. Upon the arrival of a host request, the search to determine whether the target buffer exists is performed in the same way as CFLRU, typically using hash lists, and has a time complexity of O(1). If the target buffer is found (buffer hit), the found buffer is moved to the head position of one of the clean LRU lists, the dirty LRU list, or the SC dirty LRU list. Each list is implemented as a doubly linked list, so the time complexity of this operation is O(1). If the buffer is not found (buffer miss), the tail buffer of the clean or dirty LRU list is replaced, and the new buffer is moved to the head of the clean or dirty LRU list. This time complexity is also O(1). As a result, the time complexity of processing foreground I/O requests is O(1).

On the other hand, the worst-case time complexity of the early flush performed during idle time is O(N). When performing the early flush, the tail buffer of the dirty LRU list is the target. If the write count of the tail buffer exceeds 1, the tail buffer is moved to the SC dirty LRU list, and the next tail buffer becomes a new candidate. Therefore, in the worst-case scenario, the time complexity is O(N) since all buffers in the dirty LRU list must be checked. This is an additional computational overhead compared to the CFLRU policy. However, since the early flush is performed when there are no foreground I/O requests, this overhead does not significantly affect the response time of foreground I/O requests.

Meanwhile, the proposed policy must store write count and location information in each buffer. When a buffer exists in the clean LRU list, its location can be identified from the buffer state (clean) information. Therefore, the location information can be expressed with 1 or 2 bits, depending on the implementation. The minimum value for the write count is 0, and the maximum is unlimited, but a suitable upper limit can be set. For example, the write count can be represented by 6 bits. In this case, the location information and the write count can be stored as one variable using an additional 1 byte of space. If the total capacity of the buffer cache is 1 GB, and each buffer is 8 KB in size, the additional memory overhead is 128 KB, which is not excessive. Even if the location information and the write count are stored in a 4-byte variable, the additional memory overhead is 496 KB, which is still not excessive compared to the 1 GB capacity. This additional memory overhead is also considered when evaluating performance. That is, the buffer cache capacity of the proposed policy is set to be smaller than the other policies by the memory overhead. 

## 4. Performance Evaluation

### 4.1. Configuration Setup

The performance of the CFLRU-EF-SC policy was evaluated using the SSDSim simulator [9]. We implemented three policies, CFLRU [14], CFLRU-EF [25], and CFLRU-EF-SC, in the simulator. To model the target SSD, we referred to a recent SSD performance study [3] and 3D NAND flash specifications [33,34]. The target SSD consists of four NAND flash chips connected by four parallel channels. Each chip has two dies, and each die has two planes. Each plane contains 911 NAND blocks, and each block has 575 NAND pages. The NAND page size is 8 KB. The total storage capacity is about 64 GB. The latencies for page read, page write, and block erase operations are 45 µs, 700 µs, and 4 ms, respectively. Sending a byte of data over the channel takes 25 ns, and the memory access time is 15 ns per word. The page mapping FTL [29] is used, and dynamic page allocation is employed, meaning that the target NAND page on which data are written is dynamically determined based on the channel and chip state [9].

The capacity of the SSD internal buffer varied from 32 MB to 1 GB, with each buffer having the same size as the NAND page size. It should be noted that the CFLRU-EF-SC policy incurs additional memory overhead compared to CFLRU and CFLRU-EF. Therefore, we set the buffer cache capacity of the proposed policy to be smaller, assuming that an additional 4 bytes are used per buffer. For example, when the cache capacity is 1 GB, CFLRU and CFLRU-EF have about 128K buffers, but CFLRU-EF-SC has 128K-64 buffers, which is 64 buffers fewer. This capacity setting applies to all simulation environments.

We used Microsoft Research Cambridge traces (camresweba03-lvm0, hm0, prn0, proj0, and prxy0) [35], which are representative server workloads, as the input traces. These traces are logs of block input/output requests over a one-week period. However, since each trace is only a partial log, there are numerous read requests for unwritten sectors, which the SSDSim simulator is unable to accurately measure. To address this issue, we wrote those sectors to the SSD before conducting each performance measurement. Once the SSD was initialized and there were no read requests for unwritten sectors, we conducted the performance evaluation.

The detailed attributes of the traces are shown in Table 1. “accessed logical space” refers to the amount of logical storage space that is requested to be written at least once. For instance, in the hm0 trace, a total of 20.48 GB was written to an address space of 2.50 GB, while in the prxy0 trace, a total of 53.80 GB was written to an address space of 0.96 GB. This indicates that the prxy0 trace exhibits a higher degree of spatial locality compared to the hm0 trace. 

Recently, virtual machines have enabled one physical server to support multiple virtual servers. Therefore, we conducted performance evaluations for workloads with a mixture of individual traces, as well as individual traces. To generate mixed workloads, we divided the entire logical address space into individual virtual servers. For example, hm0prn0 is a combination of hm0 and prn0 traces, where it is assumed that hm0 is located at the bottom of the logical address space, while prn0 is located at the top. We added the starting sector position of the space where prn0 is located to the sector numbers in the original prn0 trace to combine it with the hm0 trace. The resulting requests were then sorted in ascending order according to their issue time. Similarly, hm0prn0proj0prxy0 is a combination of four traces and created in the same way as hm0prn0.

### 4.2. Results in an Environment without Periodic Flushing 

This section presents the performance evaluation results of using NVRAM or battery-packed DRAM as the internal buffer of the SSD. In this environment, dirty buffers are flushed only when they are replaced or selected as the target for the early flush. 

#### 4.2.1. Response Time

Figure 4a–f depict the average response time of I/O requests for each trace. The X axis of each figure represents the buffer cache capacity, which varies from 32 to 1024 MB. The Y axis represents the average response time of the I/O requests, in units of µs. 

The results show the following. First, performing the early flush is highly effective in reducing the average response time, especially in more write-intensive traces. In all traces, both the CFLRU-EF-SC and the CFLRU-EF policies show significantly lower response times than the CFLRU policy, regardless of cache capacity. For instance, the average response time of the CFLRU-EF-SC policy is 0.70–0.94x in that of the CFLRU in camresweba03-lvm01, 0.24–0.38x in hm0, 0.08–0.29x in prn0, 0.14–0.79x in proj0, 0.03–1.00x in prxy0, and 0.15–0.22x in the hm0prn0projprxy0 trace. The degree of improvement tends to decrease as the ratio of read requests increases because a read-intensive workload such as camresweba03-lvm01generates more clean buffers, resulting in a decrease in the benefit of the early flush.

Second, among the early flush policies, the CFLRU-EF-SC shows better average response time than the CFLRU-EF in most cases. Compared with the CFLRU-EF, the response time of the CFLRU-EF-SC is shortened by −1.6–8.4% (camresweba03-lvm01), by −6.2–23.7% (hm0), by −4.9–10.4% (prn0), by −9.6–20.2% (proj0), by −4.5–28.5% (prxy0), and by 6.1–13.3% (hm0prn0proj0prxy0). The main purpose of the CFLRU-EF-SC is to reduce the increase in the total count of NAND write operations, which is a side effect of the CFLRU-EF, but it also improves the average response time in most cases.

Meanwhile, in a server environment, it is crucial to improve not only the average performance but also the worst performance. Figure 5 compares the tail response time, which is the average response time of the 1% of requests with the longest response time among all requests. The Y axis represents the tail response time of the I/O request, in units of ms. Since the results of the traces are similar, only the results of the camresweba03-lvm0 (read-intensive) and hm0prn0proj0prxy0 (mixed write-intensive) traces are presented here. 

The results indicate that the early flush is generally effective in reducing the tail response time. In both traces, the CFLRU-EF-SC and CFLRU-EF policies show significantly lower tail response times than the CFLRU, regardless of cache capacity. For example, the CFLRU-EF-SC policy reduces the tail response time of the CFLRU to 0.15–0.59x in the camresweba03-lvm0 trace and 0.26–0.77x in the hm0prn0projprxy0 trace. Among the early flush policies, the CFLRU-EF-SC generally exhibits slightly shorter response times. The improvement reaches up to 20.1% in camresweba03-lvm0 and up to 32.0% in hm0prn0proj0prxy0.

#### 4.2.2. The Total Count of NAND Writes

Figure 6 illustrates the total NAND write counts of the early flush policies compared to the CFLRU, with the relative ratio scaled by setting the write count of the CFLRU as 1 for easy comparison. 

The results indicate that giving a second chance to dirty buffers with rewrite history during the early flush is highly effective in reducing the count of NAND write operations. The CFLRU-EF-SC policy resulted in lower NAND write counts than the CFLRU-EF policy in the most configurations. Specifically, the decrease ratio of the CFLRU-EF-SC policy compared to the CFLRU-EF policy ranged from 0.0% to 4.3% in camresweba03-lvm01, −2.0% to 12.1% in hm0, 1.7% to 4.8% in prn0, 2.3% to 21.6% in proj0, −1.6% to 54.3% in prxy0, and 4.2% to 18.0% in hm0prn0proj0prxy0. 

Even compared to the CFLRU policy, the CFLRU-EF-SC policy outperformed in most configurations. The relative ratio of the CFLRU-EF-SC policy compared to the CFLRU policy ranged from 0.93 to 0.99 in camresweba03-lvm01, 0.85 to 0.99 in hm0, 0.95 to 1.02 in prn0, 0.80 to 0.93 in proj0, 0.46 to 1.02 in prxy0, and 0.83 to 0.96 in hm0prn0proj0prxy0. The maximum increase rate was about 2% in the prxy0 trace. In the mixed trace, hm0prn0proj0prxy0, the maximum decrease rate was 17.5%. 

The CFLRU-EF-SC policy exhibits relatively good performance, especially for the proj0 and prxy0 traces, when the cache capacity is set to 64 MB. Table 1 shows that these traces exhibit a high concentration of read/write requests in a relatively smaller logical storage space. Consequently, the cache hit ratio shows significant variability depending on the efficiency of managing the limited cache space. The CFLRU-EF-SC policy offers a second chance to the buffers with rewrite history, which delays their replacement and results in a higher cache hit ratio. This increased hit ratio leads to a reduction in the number of NAND write operations. As the cache capacity increases, the difference in hit ratio between policies diminishes. Consequently, the relative advantage of the CFLRU-EF-SC policy also diminishes. This trend is particularly noticeable in the prxy0 trace, which has a total logical space of only 0.96 GB. When the cache capacity is 64 MB, the CFLRU-EF-SC policy reduces write operations to 0.46x compared to the CFLRU policy by providing a second chance to the dirty buffers with rewrite history and thereby achieving a higher hit ratio. However, if the cache capacity is 256 GB or more, the CFLRU-EF-SC policy loses its benefit because cache misses other than cold misses occur rarely. In this case, only the disadvantage of the early flush is highlighted, resulting in up to 1.8% more NAND write operations compared to CFLRU.

Meanwhile, the CFLRU-EF generally exhibits a higher NAND write count compared to the CFLRU. For example, in the hm0prn0proj0prxy0 trace, the NAND write count of the CFLRU-EF increases by up to 2.2%. 

#### 4.2.3. Hit Ratio

As described above, one of the main reasons why the CFLRU-EF-SC exhibits fewer NAND writes than the CFLRU is that it gives the buffers with rewrite history a second chance, which delays their replacement and leads to a higher cache hit ratio. Figure 7 compares the cache hit ratios of the policies in the camresweba03-lvm01, prn0, proj0, and prxy0 traces. 

The results indicate that the hit ratios of all policies increase as the cache capacity increases, reaching approximately 68% for camresweba03-lvm01, 69% for prn0, 95% for proj0, and 99% for prxy0. While the difference in hit ratio between the policies is generally small, the CFLRU-EF-SC policy has a higher hit ratio than the other policies in most cases. In particular, the improvement is more noticeable in the proj0 and prxy0 traces, where read/write requests are concentrated to relatively small logical space, as shown in Table 1. Notably, when cache capacity is 64 MB, the CFLRU-EF-SC policy achieves a higher hit ratio than the CFLRU by 58.7% in proj0 and by 17.6% in prxy0. In other words, the number of cache misses are reduced by 20.5% in proj0 and by 53.5% in prxy0, which leads to the reduction in the NAND write operations, as shown in Figure 6. By giving a second chance to the dirty buffers with the rewrite history, their replacement is delayed, while the dirty buffers without the rewrite history are replaced relatively earlier, leading to an improvement in the cache hit ratio.

### 4.3. Results in an Environment with Periodic Flushing

When using volatile memory, such as DRAM, as the internal buffer of an SSD, dirty buffers must be forcibly flushed to NAND upon request from the operating system or application programs. Since dirty buffers are frequently switched to clean buffers by the forced flush, the effect of early flushing is expected to decrease. In this section, we present the performance evaluation results of the CFLRU-EF-SC policy in an environment where forced flushing is performed every 30 s. Since the evaluation results between traces were similar, we present only the results for the hm0prn0proj0prxy0 trace.

The results show the following. First, as shown in Figure 8a, the early flush is effective in reducing the average response time even in the environment where periodic forced flushing is performed. The improvement is particularly noticeable when the cache capacity is small. For example, when the capacity is 32 MB, the average response time of the CFLRU-EF-SC is only 0.27 times that of the CFLRU. On the other hand, when the capacity is 1 GB, the response times of all policies are almost the same, and the effect of performing the early flush becomes insignificant. Among the early flush policies, the CFLRU-EF-SC achieves a shorter average response time than the CFLRU-EF in most cases, although the difference is not large. 

Second, the early flush also improves the tail response time (Figure 8b). For instance, when the capacity is 256 MB, the tail response time of the CFLRU-EF-SC is only 0.17 times that of the CFLRU. Compared to the CFLRU-EF, the CFLRU-EF-SC shortens the tail response time in most cases.

Third, the increase in the count of NAND write operations, a side effect of the early flush, is insignificant, as shown in Figure 8c. This is because flushing dirty buffers in advance also occurs in the CFLRU, which does not perform the early flush. Therefore, the increase in write operations compared to the CFLRU is less than 1%, even in the CFLRU-EF. Furthermore, although the difference is insignificant, the CFLRU-EF-SC achieves fewer NAND write operations than the CFLRU-EF. In most cases, the number of NAND writes is lower than that of the CFLRU.

Finally, the difference in hit ratio is not significant (Figure 8d). This is because the operations of the CFLRU and the CFLRU-EF become similar due to the forced flush. The CFLRU-EF-SC has a slightly higher hit ratio, especially when the cache capacity is small.

In conclusion, early flushing is effective because it contributes to reducing the average and tail response times without significantly increasing the count of NAND write operations, even in an environment where dirty buffers are forced to be flushed periodically. Furthermore, the CFLRU-EF-SC is superior to the CFLRU-EF since it delivers a shorter response time while reducing the count of NAND write operations.

## 5. Conclusions

In this study, we proposed the CFLRU-EF-SC policy to address the issue of increasing the count of NAND write operations caused by the early flush. The policy evaluates the rewrite probability of candidate dirty buffers during the early flush and delays flushing by providing a second chance if the candidate has a high rewrite probability. Conversely, candidate dirty buffers that are unlikely to be rewritten are immediately flushed. Performance evaluation results showed that the proposed policy reduced NAND write operations by up to 18.0% compared to the existing early flush policy in the mixed trace. Notably, higher cache hit ratios achieved by the proposed policy led to fewer NAND write operations than the CFLRU policy in several configurations without compromising I/O request response time. In most configuration settings, the proposed policy achieved slightly shorter average and tail response times than the existing early flush policy. In mixed trace, compared to the CFLRU policy, the average response time was reduced by up to 0.15x in the environment without forced flushing and up to 0.27x in the environment with periodic forced flushing. The overheads of the proposed policy were not excessive, with an additional memory overhead of 1 to 4 bytes per buffer, and the performance evaluation was conducted considering this memory overhead. The cache capacity of the proposed policy was set smaller than that of the other policies by the amount of memory overhead, and the response time was improved. The time complexity of operating the policy was O(1) for buffer hits and misses. Because the early flush is performed in the background during idle time, it is unlikely that this overhead unduly damages the response time of foreground I/O requests. Nonetheless, this is an issue to be addressed in the future.

## Figures and Tables

**Figure 1 micromachines-14-00796-f001:**
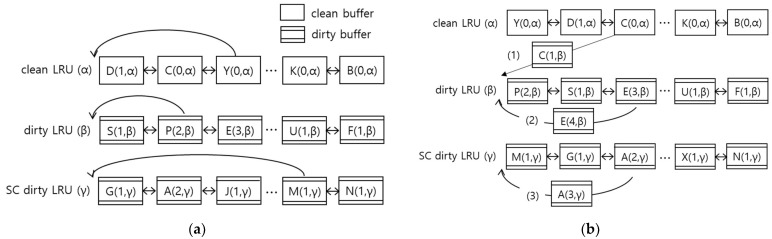
Main operations of the CFLRU-ER-SC policy. (**a**) Read hit; (**b**) write hit.

**Figure 2 micromachines-14-00796-f002:**
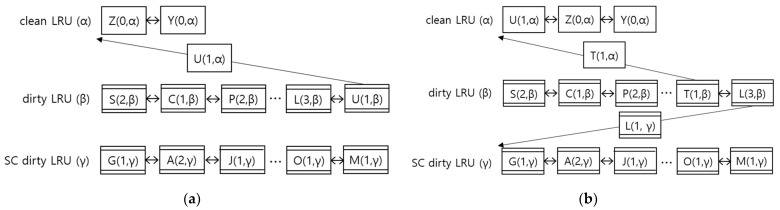
Operations of the early flush. (**a**) A case of an immediate early flush; (**b**) a case given a second chance.

**Figure 3 micromachines-14-00796-f003:**
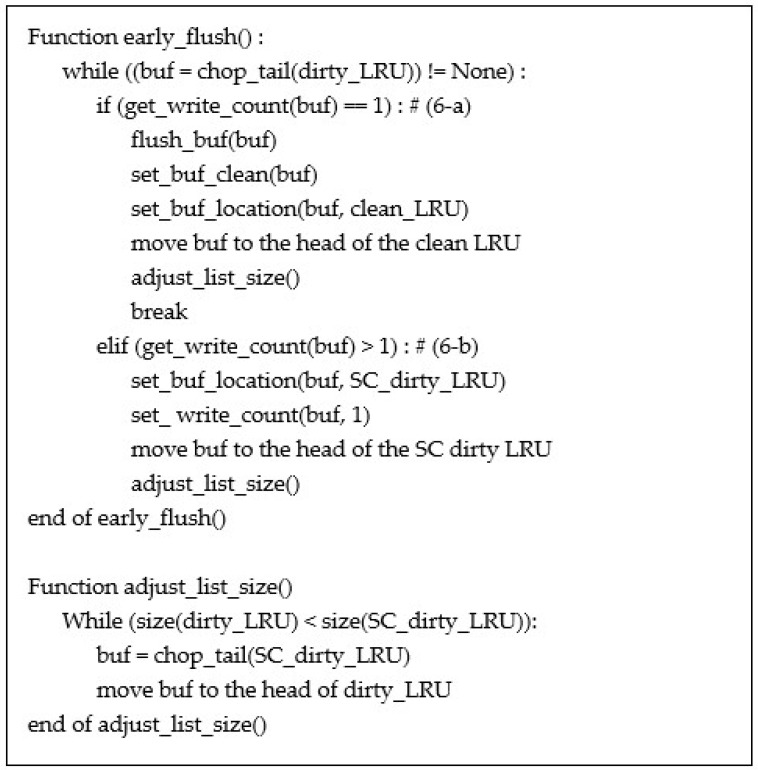
The pseudo code of the early flush.

**Figure 4 micromachines-14-00796-f004:**
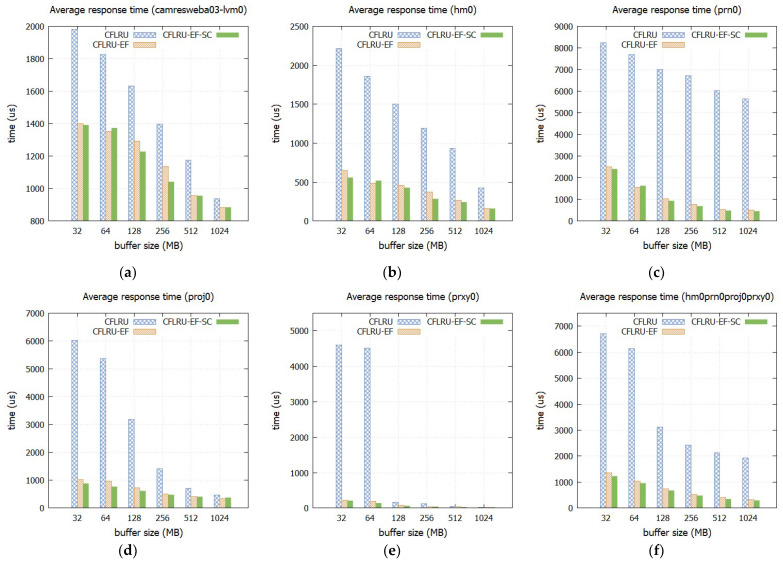
Average response time. (**a**) camresweba03-lvm0; (**b**) hm0; (**c**) prn0; (**d**) proj0; (**e**) prxy0; (**f**) hm0prn0proj0prxy0.

**Figure 5 micromachines-14-00796-f005:**
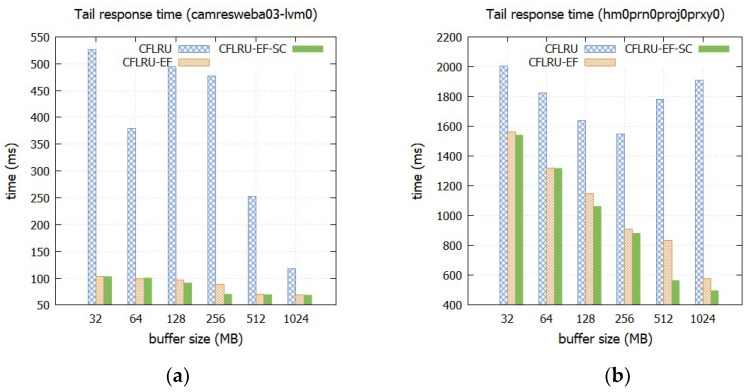
Tail response time. (**a**) camresweba03-lvm0; (**b**) hm0prn0proj0prxy0.

**Figure 6 micromachines-14-00796-f006:**
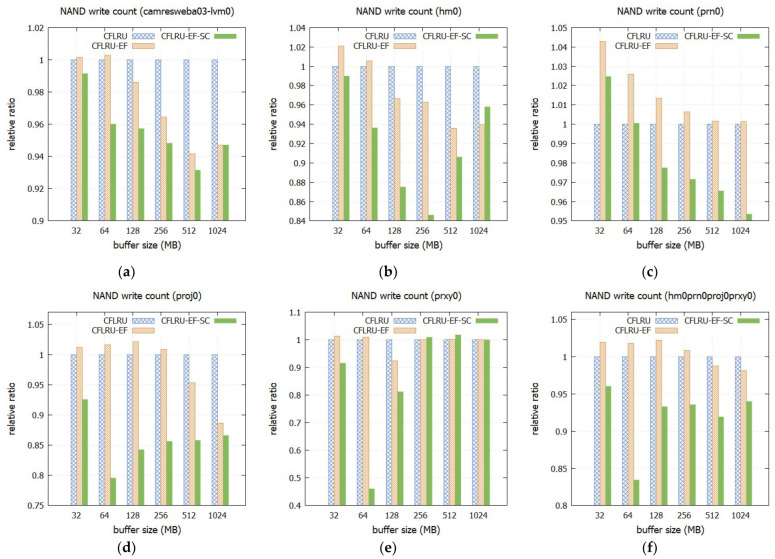
The number of NAND write operations. (**a**) camresweba03-lvm0; (**b**) hm0; (**c**) prn0; (**d**) proj0; (**e**) prxy0; (**f**) hm0prn0proj0prxy0.

**Figure 7 micromachines-14-00796-f007:**
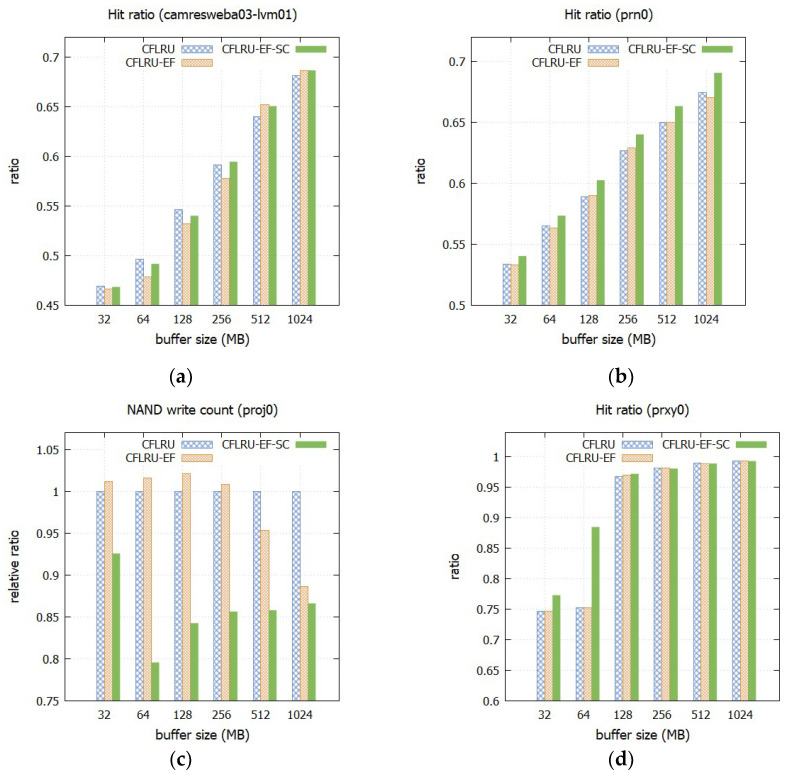
Cache hit ratio. (**a**) camresweba03-lvm01; (**b**) prn0; (**c**) proj0; (**d**) prxy0.

**Figure 8 micromachines-14-00796-f008:**
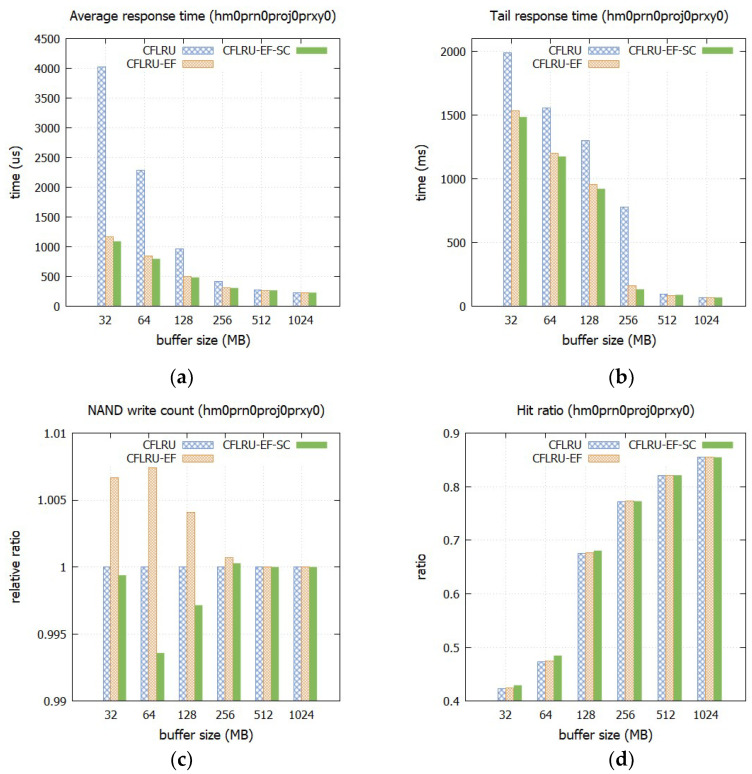
Results with periodic flushing. (**a**) average response time; (**b**) tail response time; (**c**) NAND write count; (**d**) hit ratio.

**Table 1 micromachines-14-00796-t001:** MSRC trace attributes.

Trace	Accessed Logical Space (GB)	Total Read/Written Bytes (GB)	Read/Write Ratio (%)	Avg. Read Req. Size (Sectors)	Avg. Write Req. Size (Sectors)	Avg. Inter-arrival Time (ms)
Camresweba 03-lvm0	7.30	17.35/11.67	59.8/40.2	60.0	17.2	0.30
hm0	2.50	9.96/20.48	32.7/67.3	14.7	16.7	0.15
prn0	14.90	13.12/45.97	22.2/77.8	45.7	19.3	0.11
proj0	3.27	8.97/144.27	5.9/94.1	35.7	81.8	0.14
prxy0	0.96	3.05/53.80	5.4/94.6	16.7	9.3	0.05
hm0prn0 proj0prxy0	21.63	35.10/264.51	11.7/88.3	25.1	23.7	0.02

## Data Availability

Not applicable.

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
