# Peer review of "Early Dirty Buffer Flush with Second Chance for SSDs"

_micromachines, 2023, doi:10.3390/mi14040796_

Round 1

Reviewer 1 Report

The manuscript tackles interesting performance issue for SSDs.

The author contributes incremental improvement of their previous work.

The manuscript explains the proposed policy well, and the policy is systemically well defined.

Several improvement are needed in the Peroformance Evaluation section.

 - It needs to describe how to merge multiple workload traces.

 - It needs to explain reasons why there is such a large performance improvement in proxy0 trace. Is it related to the workload characteristics? 

 - It'd be better to include result bars of CFLRU in the write count graph.  

Reviewer 2 Report

The author has done some research to optimize early dirty buffer flushing. Which dirty buffer is flushed depends on whether it has a high possibility of rewriting. Specifically, dirty buffer with high rewriting probability will delay flushing. The experiment shows that selective early flush policy reduces NAND write operations by up to 20.1% compared to the existing early flush policy in the mixed trace. 

Shortage

The innovation of this article is not enough, and the description of contributions is too short.

The article does not explain why the write count difference between the two cache flushing strategies is so large when the buffer size is 128MB in Figure 6(c~f).

Detailed comments:

The idea of the article is easy. Reducing NAND write operations by optimizing dirty cache flushing strategy.

Here are some suggestions for improving the article:

1. Adding some innovations.

2. The size of the pictures in the experimental part should be appropriately reduced, and the arrangement should be compact, and the focus should be on innovation and contribution.

3. For the selected traces, both are write-intensive traces. So you also should give some read-intensive traces. Furthermore, you also should give some detailed attributes for the mixed trace.

Round 2

Reviewer 2 Report

My previous comments have been mostly addressed. Thanks for the revision. However, for the read-intensive traces, read ratio is higher than 50% (in your current version, there are no these traces). So you still give these traces, and show its effectiveness under these traces. 

Round 3

Reviewer 2 Report

The authors have clarified all my questions. I do not have any further problems/questions.